# Teaching Complex Introductory Concepts in a Sophomore Circuits Course: A Descriptive Case Study

Nicole Pitterson

Department of Engineering Education, Virginia Tech, Blacksburg, VA 24061, USA; npitters@vt.edu

**Abstract:** This descriptive case study explores the teaching and learning of complex introductory circuit concepts in a compulsory sophomore circuits' course. The study investigates the instructional strategies employed by the instructor to facilitate students' understanding of intricate circuit phenomena. Data were collected through classroom observations, interviews with the instructor, and an analysis of the course documents. The findings shed light on the challenges encountered by students when grappling with introductory circuit concepts, the effectiveness of different instructional methods, and implications for curriculum design and pedagogical approaches in electrical engineering education. Specifically, the instructors reported students' prior knowledge, the nature of the content, and the structure of the course itself as some of the main features that impact students' overall learning of the content. The study highlights the importance of providing targeted support and scaffolding to students, promoting active learning strategies, and incorporating practical applications to enhance the comprehension of introductory circuit concepts in sophomore-level electrical engineering courses.

**Keywords:** instructional methods; student engagement; curriculum design; pedagogical approaches





## 1. Introduction

Electrical engineering introductory circuit courses are the first context in which students are exposed to simple, steady-state, or direct current (DC) to complex, transient, or alternating current (AC) circuit theory and concepts. While students are usually taught basic circuit concepts in physics classes, introductory classes tend to go into more depth, as these classes form the basis for a specialization in electrical engineering [1]. Research has indicated that, despite the addition and application of active learning strategies to the teaching process, students still experience difficulties learning these complex circuit concepts [2–5]. Previous studies have focused on students' epistemological and ontological beliefs about the concept of electricity and how they foster misconceptions [5–11]. However, much of the work that has been performed on circuit concepts has focused on the difficulty associated with the learning of the concept based primarily on students understanding [12–14] and, to a greater extent, on basic or direct current (DC) circuits [15–19].

In addition, an area of concentration has been on the impact of learning environment design and the nature of instruction on undergraduate engineering education [20–22]. Nevertheless, there are a lack of studies on the design of learning environments in terms of the decisions made about the teaching of complex circuit concepts and how these decisions are influenced by students' perceived prior knowledge. This is an important area to be researched, as it helps to uncover the relationship between the techniques used to express information about these circuit concepts and possible barriers to students' understanding.

Work in this space can explore the alignment of content, the way the content is taught, and what concepts are emphasized as important for conceptual understanding, as well as how great a role students' prior knowledge plays in the dissemination of this knowledge. This study focuses on the teaching of introductory circuit concepts in a compulsory, sophomore-level circuits course using a descriptive case study approach. The research questions that guided this study were:

a. How are introductory circuit concepts taught to students enrolled in a compulsory sophomore level course?
b. What factors, if any, influence instructors' decisions regarding how to teach circuit concepts to students?

## 2. Background

There are several factors that influence how students learn and the decision-making process that instructors undergo when designing their courses and delivering content. In this section, the main factors, supported by the previous literature, are discussed. The theoretical framework guiding this work is also discussed at the end of the section.

### 2.1. The Role of Students' Prior Knowledge

Engineering students' ability to learn introductory concepts is very important for their success in becoming experts in their respective disciplines or areas of study. More specifically, "to develop competence in an area of inquiry students must have a deep foundation of factual knowledge, understand facts and ideas in the context of a conceptual framework and organize knowledge in ways that facilitate retrieval and application" ([23], p. 16). According to [24], the process of learning is characterized "in terms of comprehension, skill acquisition, both" (p. 440). These guiding principles can be applied explicitly to the introductory circuit course studied in this paper that all engineering majors are required to pursue. Therefore, the practice of learning in introductory classes has implications for the materials presented to students and the decisions surrounding the style in which they are presented. In this section of the paper, the existing literature will be synthesized to explore what counts as foundational knowledge in relation to understanding circuit concepts and how this knowledge is typically communicated in engineering learning environments.

Engineering practice, as categorized by [25], consists of three components:

1. Engineering as problem solving, considering the systematic process that engineers use to define and resolve problems.
2. Engineering as knowledge, considering the specialized knowledge that enables and fuels the process.
3. Engineering as the integration of process and knowledge (p. 429).

In keeping with these three core areas, the root of electrical engineering expertise can be classified as a working knowledge of basic to complex circuit concepts which is transferred from course to course, an advanced mathematical understanding, and the combination of content knowledge and mathematical skills which develops the ability to identify and solve unknown circuit conditions.

At the surface level, the heart of electrical engineering knowledge can be characterized by the ability to identify following conditions:

a. The three basic circuit configurations: series, parallel, and series–parallel,
b. The four dominant variables: voltage, current, resistance, and power,
c. The four main components of electric circuits: source, control, load, and conductors, and
d. An understanding of how all these factors interact to create the desired circuit operation.

Research, however, has indicated that students tend to have difficulty understanding these very basic concepts, which then becomes problematic when more complex concepts are introduced [11,13,14,26]. The work of Shaffer and McDonald [27] has been cited as one of the hallmarks of the research conducted on investigating the difficulties students experience when learning direct current (DC) circuit concepts. In this study, the authors sought to investigate the difficulties students experience when learning simple electric circuits that relate to the four conditions discussed previously. In addition, ref. [28] used various studies expanded on the categories and sub-categories summarized in the table above to include:

a.　　An inability to handle simultaneous change of variable (p. 37).
b.　　An inadequate use and misuse of analogies (p. 47).
c.　　A fear of qualitative reasoning—the mechanical use of formulas (p. 49).

Similarly, Bernhard and Carstensen [29] and Streveler et al. [3] reported that a basic understating of the relationship among various electrical quantities is an important area of difficulty for students. Students tend to have difficulty envisioning quantities such as voltage, current, and resistance acting interchangeably in a circuit, yet still performing their own circuit task toward the holistic operation of the circuit [30,31]. In each case, a recommendation has been made for the use of specific instructional strategies possessing the ability to help students to overcome these difficulties. This is based on the premise that students not only learn these basic introductory concepts, but are able to apply them to more complex contexts such as other courses and the world of engineering practice. This involves the ability to transfer knowledge. However, it has been discussed that "one's existing knowledge can also make it difficult to learn new information" ([20], p. 70). The transfer of knowledge is highly dependent on mastery of the initial information, which involves a deep conceptual understanding rather than the memorization of facts. To achieve this deep conceptual understanding and the ability to apply what is being taught, sufficient time to process and explore related connections to other concepts as exposure to various means of representation is a necessity [20,32].

### 2.2. The Nature of Introductory Courses

The primary goal of introductory courses is to ensure that students develop a foundational knowledge that can be transferred to more complex concepts as they progress toward degree completion and beyond [21,33]. Thus, researchers recommend that learning environments should support active engagement and guide students towards the acquisition of self-regulated processes [21,34]. In such a setting, students are encouraged to construct their own knowledge and skills in learning these concepts through actively navigating their role as learners [35,36]. Any investigation of students' ability to reflect on their prior knowledge and how it impacts their explanation of concepts long after they have exited the learning environment highlights the decisions made by instructors about which concepts to reinforce as significant.

Since knowledge acquisition is a key role in introductory courses, one of the main pedagogical principles often employed is problem solving through repeated practice [37]. Problem solving in introductory circuits courses emphasizes the acquisition of the fundamental theories and mathematical techniques required to analyze and design circuits [19,38]. The role of problem solving should not be understated, as this provides a means by which students can solve problems in a controlled, yet repeated way that further influences their long-term understanding of the concepts being taught. According to [37], deliberate repeated practice has significant benefits for learning fundamental concepts that can then be transferred to future and more advanced courses.

### 2.3. The Role of Mathematical Thinking

Mathematical thinking and being able to apply theoretical mathematical principles are critical to learning and analyzing circuit concepts due to the high level of abstraction that is associated with this aspect of electrical engineering. According to Schoenfeld [39], an advanced level of mathematical understanding is paramount to learning complex concepts as "the tools of mathematics are abstraction, symbolic representation and symbolic manipulation" (p. 3). These are the principles upon which circuit analysis and understanding are built. The role of advanced mathematical thinking is evident in the level of math that is required in most cases before students can enroll in circuit analysis courses [40]. For example, the work of Faulkner et al. [41] espoused that pursuing an engineering degree entails enrolling in and successfully completing a series of mathematics courses. Typically, this sequence includes multiple semesters covering topics such as calculus, linear algebra, and differential equations. More specifically, readiness for and successful progression

through calculus are often identified as strong indicators of one's likelihood to graduate with an engineering degree [42,43]. Consequently, in the realm of learning circuits, advanced mathematical thinking can aid how students engage in problem solving, critical thinking, abstraction, logical reasoning, and pattern recognition, which are necessary skills for circuit analysis [41,44].

*2.4. Guiding Framework—Pedagogical Content Knowledge (PCK)*

Pedagogical content knowledge (PCK) as a framework is used in research to highlight how the knowledge and beliefs held by instructors influence their classroom practice. This framework posits that, as instructors blend their own knowledge about specific content and their experiences, they tend to present content to their students in the form they believe best enables learning [45]. In addition, instructors use their PCK to determine: (1) what concepts are important for emphasis, (2) the teaching strategies that are most effective for teaching specific topics and, (3) the learning activities necessary to foster conceptual understanding [46]. Though PCK has its roots in scientific education and is often used as a construct for measuring a science teacher's use of their own knowledge to become effective in teaching, PCK can also be used as a guiding principle for data collection and analysis in studies aimed at investigating the nature of scientific content and student learning. For the purposes of this study, the five components of PCK, as discussed by Magnusson, Krajcik, and Borko ([44], p. 97), were used to guide the collection of data for this study. These are:

1.  Orientations toward science learning: this involves daily instructional decisions regarding class objectives and content, student engagement, and the use of curricular materials (p. 97).
2.  Knowledge and beliefs about science curriculum: this involves how information about the goals of the class is communicated to the students over the duration of the course, as well as the activities and materials used in achieving these goals (p. 104).
3.  Knowledge and beliefs about students' understanding of specific science topics: this involves the prerequisite knowledge and skills students are required to have, how teachers incorporate individual student ability in the dissemination of class activities, and what concepts students find difficult to understand (p. 105).
4.  Knowledge and beliefs about assessment in science: this involves the decisions made about the appropriate means for assessing student learning, such as approaches, activities, or specific procedures (p. 109).
5.  Knowledge and beliefs about instructional strategies for teaching science: this involves the various approaches used to represent scientific concepts and principles in a manner that best facilitates student learning.

This framework was used as it provides the opportunity to examine the decisions made by professors relating to how the content of a course is taught to the students, the strategies used for student engagement, and how students perceive difficulty in understanding is addressed.

## 3. Methodology

A descriptive case study is typically used to describe a phenomenon and the context in which it occurs [47,48]. In this type of research, the intent is to highlight the overarching connections among different sources of data pertaining to the phenomenon under investigation [49]. The main benefit of descriptive case studies lies within their ability to draw data from many sources, with each source being of equal importance in providing in-depth information relevant to the topic being studied [48]. In addition, findings from a descriptive case study tend to have implications that can be applicable to other cases or fields of study [49]. The study was approved and guided by the following institutional review principles under study no. 17-1106.

### 3.1. Context

Linear Circuit Analysis I is an introductory three-credit circuit course compulsory for all undergraduate engineering majors and is a core course for electrical engineering majors at a large Midwestern University in the US. This course is usually taken by sophomore engineering students. The pre-requisites for this course are an engineering design-based course, introductory physics, and three semesters of calculus, one of which can be taken concurrently. There are usually seven (7) sections of the course, which consists of five (5) lectures and two (2) distance learning components. The course is offered every Fall, Spring, and Summer. During the Fall and Spring semesters, the class meets three days a week for one hour. The accompanying lab component for this course is compulsory only for electrical engineering majors. In this course, students interface with theoretical and practical material related to simple to complex circuits and circuit principles. Basic electrical principles such as voltage, current, resistance, and power in both DC and AC circuits frame the basis of this course. The main objectives of this course are to expose students to volt-ampere relationships and characteristics and the development of the ability to analyze first- and second-order linear resistive circuits with DC and AC sources, as well as being able to compute voltage, current, power, and impedance values. Since this case study was conducted during only one course, the objective is not to generalize across all circuit courses, but to shed some light on the decisions made about this course and how these decisions guide the design and delivery of content.

### 3.2. Case

A single descriptive case with embedded units was used for the design of this study, as well as to guide the collection of the data. In this study, the case used was the introductory circuit course, with three sections chosen as the units of analysis. This approach was chosen as "subunits often add significant opportunities for extensive analysis, enhancing the insights into the single case" ([48], p. 56). To facilitate this in-depth analysis, the same types of data were collected in each unit, and each data set was first analyzed separately and then collectively.

### 3.3. Participants

The participants were the professors who instructed the sessions chosen for the study. The sessions used were selected from the results of two pilot studies. One pilot study was conducted a semester before the actual data collection period, while the other was conducted during the semester of data collection, but earlier in the semester. From the pilot studies, the unique strategies used by the professors and the way they engaged the students were noted. The times of the class periods were also a determining factor. For example, since all the classes were scheduled for the same days of the week, it was important that there was at least a 2-hour break between sessions. This was necessary so that the author had enough time to reflect on the previous class and write an analytical memo before conducting another observation. Additionally, other unique features about the three sections from which the data were collected were the lengths of experience of each professor and the sizes of the classes. The professors' experience ranged from over eight years to one year of teaching the course. In relation to class size, the three sections ranged from large (over 150 students enrolled) to relatively small (60 students enrolled).

### 3.4. Data Collection

The use of multiple sources of data helped the researcher to triangulate their findings, provides supporting evidence for the propositions made about the case, and strengthens the value of the case study [50]. In this study, the data sources selected were aimed at providing the author with data points aimed at collecting different types of information about the course. Multiple data sources facilitate the development of a holistic understanding of the case under study [47]. Using multiple data sources which are analyzed separately and then collectively adds strength to the findings of the research, as various strands of

data are interwoven together to provide a greater understanding of the case ([46], p. 554). In addition, using multiple sources of evidence for a single case has the advantage of developing converging lines of inquiry around a phenomenon [50]. These multiple sources of data were collected in each section of the course. For this paper, the focus of the data collection was primarily on how concepts were introduced and taught in general, how the transition from simple to more complex concepts was made, the role of the students in the environment, and how knowledge was communicated. The data used for this study were collected from a variety of sources, including direct classroom observations, semi-structured interviews, and course documents such as the syllabus and lectures.

### 3.5. Direct Classroom Observations

A pilot study of various observation protocols was conducted to determine which protocol was most suitable for the study. Fifteen (15) direct classroom observations, five from each section (unit), were conducted using the Teaching Dimensions Observation Protocol (TDOP) [51]. The TDOP was developed and validated by [51] through a repeated process of literature integration and classroom testing to collect information about the various instructional practices used in learning environments to communicate knowledge, integrate the use of visual and technological aids, and other types of classroom artifacts. The protocol consists of six categories, namely: teaching methods, pedagogical strategies, cognitive demand, student–teacher interactions, student engagement, and instructional technology. Within each category, a set of pre-determined codes were used to record data in two-minute intervals. In addition to the codes, detailed notes were made at each interval and an analytic memo was written following each observation.

### 3.6. Semi-Structured Interviews

Interviews as a data collection method for case studies are described as an "essential source of case study evidence because most case studies are about human affairs or behavioral events" ([50], p. 108). The three professors of the sections chosen for the study were interviewed. The interview protocol was developed by the author using a combination of the tenets of the guiding framework, the previous literature, and the data gathered from the direct course observations. The interview protocol was piloted twice with two instructors and experts in engineering education, after which, the questions were revised to ensure they were in alignment with the research questions and overall goals of the study. The interviews were intended to gain insight into the decisions made by the professors on how to teach electric circuit concepts, strategies for learning the course material, and their personal philosophy for teaching complex concepts. The structure of the interviews and the open-ended nature of the questions ensured that the researcher was able to clarify the responses as the professors answered the questions to obtain more in-depth information. Audio recordings of the interviews were taken, after which, each interview was transcribed verbatim and member checked.

### 3.7. Course Documents

Documents are social products aligned with rules and structure based on collective and organized action [46]; course documents represent an aspect of data that allow for a comparison of how students are taught about circuit concepts and their perceived role in the learning environment. The course outline and lecture notes for the period of the class that was observed for each section were collected.

### 3.8. Data Analysis

Each piece of data collected in a case study is a part of a big picture or puzzle and, as such, the most beneficial manner of analyzing these pieces of data is to show how they link to other data or the initial themes and propositions [50]. This study focused on how concepts were taught to the students enrolled in the course. Hence, the data collected centered on how concepts were introduced, how knowledge was communicated, and

how information was disseminated. Each unit was analyzed separately (within units) and then collectively (across units); this method of data analysis strengthened the findings of this study, in that it provided in-depth information about the phenomenon being studied. Descriptive coding [52,53] was used to conduct the first round of data analysis for each unit, which was then collated in themes. The themes and codes generated from the first round were used to inform the pattern coding used for the second round of data analysis conducted across all three units.

## 4. Results and Discussion

In this section, the findings from each unit are presented separately, followed by the findings across all the units. The results from the individual units' analysis respond to the first research question—how are introductory circuit concepts taught to students enrolled in a compulsory sophomore-level course?—while the results from the combined, across-unit analysis respond to the second research question—what factors, if any, influence instructors' decisions regarding how to teach circuit concepts to students?

### 4.1. Unit One Findings

#### 4.1.1. Direct Classroom Observations

Figure 1, a summary of the observations, illustrates that a high percentage of the observed sessions were conducted with a heavy reliance on classroom instructional technology and were focused on problem solving. This is evidenced by the 95% of time spent using display tablets (DT) and 92% of time spent lecturing from pre-made visuals (LVIS), as well as the 59% and 40% times for overhead projector (OP) and PowerPoint (PP) slides, respectively. In addition, 79% of the observed time was attributed to lecturing while writing (LW) and working on problems (WP). These two codes, LW and WP, are typically recorded together as a recommendation from the TDOP. The observations uncovered that the classes were highly teacher-focused, with very low interaction between students and the instructor. This was confirmed by the low percentage of student-focused dialogue, since only 12% of the observed time was student responses (SR) to instructor questions and 3% for student questions (SQ). However, student engagement was very high (VHI), at 53% of the observed time. In terms of pedagogy strategies, 56% of the observed time was spent emphasizing (EMP) concepts and equation notation.

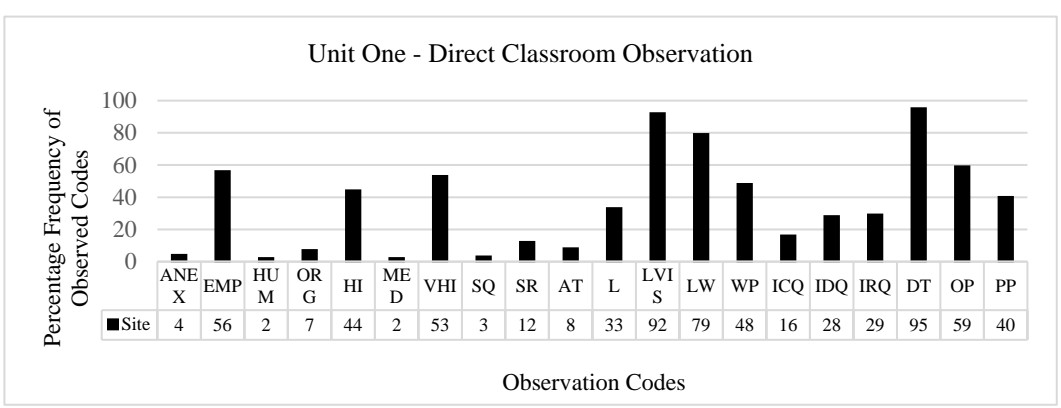

**Figure 1.** Percentage of observed codes for unit one direct classroom observations.

#### 4.1.2. Professor Interview

The recurring threads that emerged from the coding of the interviews were:

a.      Students' insufficient mathematical knowledge—this was coded when the professor would discuss how students lacked sufficient background mathematical knowledge that, in turn, influenced their ability to engage with the materials for the class successfully. See the illustrative quote below:

"[This course] requires them to apply the math skills that they either are learning or for example differential equations or some math skills that they might have not used or forgotten for example complex number analysis. So, the challenge that they need to very quickly combine the concepts that they learned, new concepts with the math skills that these are not familiar or in some cases forgotten so these are the conceptual and learning difficulties."

b.  The importance of repetitive practice was coded when the professor would discuss strategies for learning the course material.

"I tell them "this is hand memory" right? because if you practice piano your hand also has memory similar things would happen for circuit analysis if you keep practicing your brain naturally sort of recognize each question."

c.  The use of analogical and comparative examples was coded when the professor would describe their use of analogies to help students make sense of the abstract course concepts. However, the professor also cautioned that an over dependence on analogies can potentially lead to misunderstandings of the content.

"Even for, it is clear that analogies are only analogies they are not the same things so there are always subtle differences and if we emphasis too much and that will make a, sort of a hardwired link, the students they might even make mistakes down in the road and also you know some of the concepts over there are not so easy to make an analogy for…Yeah you can think about further analogies but then you run into the risk of distorting the concept to try to fit the analogy. So yeah there's a tradeoff and if you don't explain too much students don't get it, if you say this is exactly that then students they may misunderstand"

This professor also discussed some of challenges associated with helping students to achieve the course learning outcomes, such as wanting to provide students with information about the multiple strategies they can use for problem solving while dealing with other issues related to time constraints, such as having to negotiate a balance of how deep to go into content. Dealing with time constraints was an issue the professor discussed that was not common for the other sources of data. However, this factor had a significant impact on decisions such as how deeply to go into the concepts being taught or how much time could be spent ensuring students completely understood the information being presented. Overall, this professor discussed that, what was most important, in their opinion, was students' ability to apply the mathematical knowledge they had from previous courses and engaging in activities that would provide them with the practice necessary to master the concepts that were discussed as detrimental to the students' learning of the content.

### 4.1.3. Course Documents (Course Outline and Lecture Notes)

The most prominent themes that were repeated throughout the course outline and lecture notes collected for this unit were the importance of mathematical knowledge and the ability to apply the relevant problem-solving skills, as well as how repetitive practice is beneficial for deep conceptual learning. The course objective was specific to learning how to analyze different circuit configurations under varied circuit conditions. This was also present in the lecture notes and other documents students were given, such as quizzes that were used as instructional tools at the end of every topic.

### *4.2. Unit Two*

### 4.2.1. Direct Classroom Observations

Figure 2 summarizes the codes captured in the observations. The figure shows a vast use of classroom instructional technology, as is presented by the 99% of observed time in which the professor lectured from pre-made visuals (LVIS). This was also reflected in the combination codes of the use of display tablets (DT) during 98% of the observed intervals and 73% for the use of overhead projectors (OP). There was a significant proportion of the observed intervals spent lecturing while writing (LW), shown as 84%, and working on

problems (WP), shown as 57%. This supports the theme that the sessions observed were primarily instructor-focused. In 60% of the observed time, the professor would emphasize (EMP) the importance of the concepts or equations being covered for either exams or for learning new and upcoming information. Student engagement in this section ranged from very high (VHI) at 42% and high (HI) at 24% to medium (MED) at 31%. This was also evident in the low percentage of student response (SR), 6%, observed.

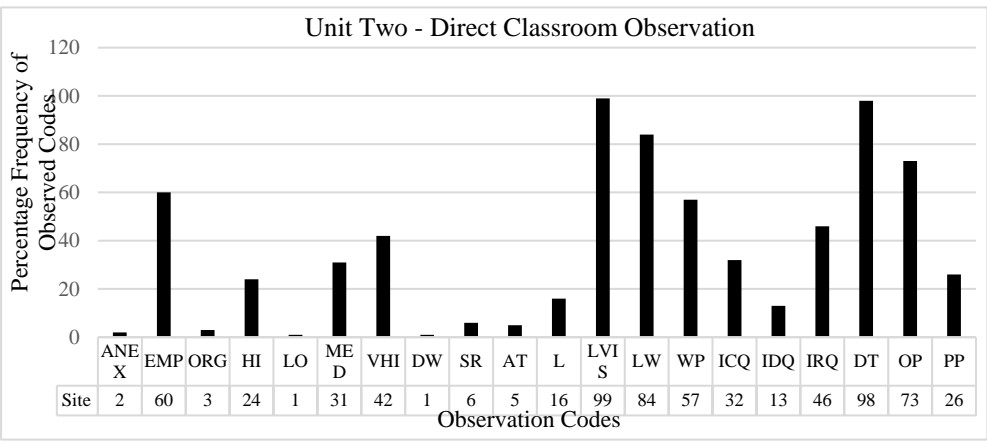

**Figure 2.** Percentage of observed codes for unit two direct classroom observations.

4.2.2. Professor Interview

The emerging themes from the observations were replicated in the interviews. In each category, the professor spoke about their perception of how these factors impacted not only how well students learned the concepts of the course, but their ability to transfer their learning to other advanced courses. For each category coded, an illustrative quote is provided.

a.  Challenges faced in delivering the content stemming from students' lack of adequate pre-requisite knowledge:

"The other thing is that a lot of them come from different mathematical backgrounds so I do see some of them struggling with the math in the course. I realize towards the later part of the lecture especially dealing with AC circuit analysis, now they need to be familiar with calculus and complex numbers a lot of them may forget or they didn't have that knowledge in hand so I try to review some related math topics with them"

b.  The importance of having students appreciate the deliberated and repetitive practice of working on problems, inside and outside of the context of the classroom:

"I think a lot of times the student can solve the problem that I give in the class, so they can follow the examples ok? But then when you're giving them a new set of problems based on the homework and exam a lot of times they kind of get lost and they don't where to start so that's why. I would say but in terms of difficulties, not much but I think it's kind of the key to being successful in this class is taking a lot of practice. I think for a lot of students that might be a little challenging to work through a lot of problems to understand."

c.  The need to provide students with the opportunity to develop the necessary problem-solving skills and the importance of transferring their knowledge to more complex contexts such as more advanced courses.

"So a lot of times I refer them to more advanced materials that, because we have a lot of open learning materials and things that if they are interested they can take a look at it or I, sometimes I tell them that these are topics covered in more higher level courses so they will understand just right now they don't have the necessary skills

to fully understand all this concept that they need a fundamental course to at least know how the circuit works then they can dig deeper into higher courses and a lot of times when I tell them that they're ok with it."

Like unit one, the use of analogical reasoning and real-life examples, as well as dealing with the imposed time constraints, also emerged from the interview and will be discussed in the across-unit findings.

### 4.2.3. Course Documents (Course Outline and Lecture Notes)

The documents collected and analyzed for this section were similar to those in unit one; this is not surprising, as the professor for unit two was being mentored by the professor of unit one. There was a strong emphasis on having the relevant pre-requisite knowledge in mathematical and physics concepts before attempting to enroll in the course, as well as the importance of repetitive practice for achieving a good grade. The integration of course outcomes and how the exams were designed was reinforced in the course documents, as well as in the interview with the professor.

### *4.3. Unit Three*
### 4.3.1. Direct Classroom Observations

In this unit, the primary means of presenting information to the students was the professor writing on the chalkboard while lecturing. As can be seen from Figure 3, the chalkboard (CB) was used for 99% of the observed time, as well as a 99% frequency of the lecturing while writing (LW) code. Contrary to the other two units, student engagement was significantly higher, with very high (VHI) engagement being observed for 97% of the time intervals. In addition, there were more observed instances of student questions (SQ) at 38% and student response (SR) at 7%. Another interesting observation was the high level of instructor comprehension questions (ICQ) posed, which was observed for 63% of the time intervals. This indicates there was more student–instructor dialogue than in the other two units observed. There was also more time spent emphasizing (EMP) concepts, at 66% of the observed intervals, which can be attributed to the increase in the dialogue with students.

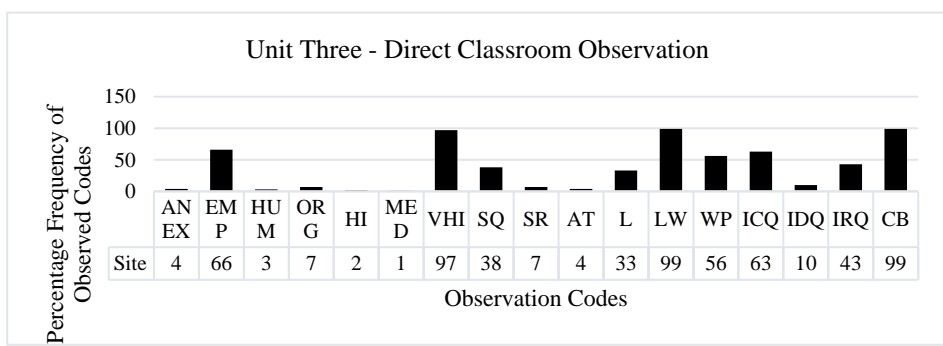

**Figure 3.** Percentage of observed codes for unit three direct classroom observations.

### 4.3.2. Professor Interview

The most common theme that emerged from this interview was how the class was designed and the information being conveyed in a manner intent on teaching students problem-solving strategies that would be applicable to more complex courses. The professor also talked specifically about how, by making their own notes, students could learn the material in a deeper way. The professor discussed their specific strategy of using the chalkboard most of the time because they wanted their students to develop the ability to not only take good notes, but to start seeing patterns across the multiple strategies used to solve the problems assigned. This finding was triangulated by the observed time spent on working on problems, deriving equations, and discussing how these equations can be applied.

"I don't give the class notes to my students in advance. I want them to write their own notes in class because I think that that actually helps them a lot if they have their own handwritten notes, while they are writing they learn if there is something they don't understand they might just put an asterisk on it or whatever and also then if I release the class notes later on which I do release them after I have taught a class, they can always see if there is something they didn't understand they can go to my class notes hopefully that will help them complete their own class notes and if there is some discrepancy between them that will draw a bulb in their heads saying "oh what's going on here? Why does my class notes say something different than what he's saying?" and that might help them think ahead and ask questions and so on, so I actually like them to get their own class notes and then look at mine if they need to."

Other common themes were related to problems with transferring knowledge, the use of analogical comparisons and real-life applications to engage students even though this was difficult, the need for repeated practice to go beyond memorization, and problematic pre-requisite knowledge.

a.  Problems with transferring knowledge were coded when the professor discussed how they made decisions about what they showcased to their students from one class session to the next.

"I try to give them a little bit, a taste of what goes beyond what they can do with these kinds of things and try to offer some complicated concepts and some difficult things which they don't really need for this class but those students who are interested in them and have the capability might feel bored with what we are doing because they are already on top of it, they have something to study beyond the basics because that's going to be useful for later on ok. When they go to the next classes there is a "oh I have already seen all of this" when they go to Fourier transform "oh I learn frequency response, I understand why that's still useful, I can decompose this into different frequencies" so I try to give them a bit more than just what the syllabus says is the minimum."

b.  The use of analogical comparisons and real-life applications to engage students was coded when the professor discussed the various strategies they implemented to engage students in class discussions. However, like the other two professors, they also cautioned against the overuse of analogies, as they can lead to misunderstandings of the concepts.

"At the beginning when I try to explain current and voltage I try to draw an analogy to water circuits for different voltage levels or different heights of water through different recipients, current is just the flow of water so yeah I try to introduce analogies as often as I can but that's, so the problem with analogies is that sometimes things don't map one to one right, so sometimes people come to me and tell me "hey you said that this was like water in two tanks" and I'm like "yeah ok but that worked for that specific aspect and for this particular aspect it doesn't work" and then they're like "oh ok" and suddenly they have built this whole castle based on that analogy which has basically fallen down because it doesn't work in this aspect. I try to strike a tradeoff between both things, I try to, for this specific analogy about the water I use it during the first week or two when I explain voltage and current and then never use it again in the class."

### 4.3.3. Course Documents (Course Outline and Lecture Notes)

Interestingly, the lecture notes for this section consisted of discussions about the concept being covered in a similar manner to the in-class teaching. Rather than mathematical symbols and notations, the students were presented with material aimed at helping them understand the nature of the concepts, followed by the mathematical principles they could use to prove interaction among variables. Though the need for relevant mathematical

pre-requisite knowledge and skills were important factors, the idea of discussing why these skills were necessary was explicitly addressed in the course outline and lecture notes.

### 4.4. Across Unit Findings

The second round of data analysis was conducted based on the findings of the individual unit descriptive coding. The pattern-coding approach was used to categorize the themes. There were five patterns that emerged across the data set. These were: the perceived characteristics of students, the perceived characteristics of content, the structured problem-solving process, student engagement, and the impact of time constraints.

The cross-unit analysis identified, on a broad scale, some of the decisions made by the professors about how to teach the content while dealing with the imposed constraints associated with the curriculum, students' prior knowledge, and what the course objectives were. These findings indicated there was a cross-section between what was an expectation of having pursued this course, the nature of the content being taught, and other impeding factors such as what knowledge and experiences the students brought with them to the learning process. In this section, how these findings align with the literature will be discussed.

### 4.4.1. Perceived Characteristics of the Students

The data collected from the three units reinforced the idea that there is a core body of knowledge students must have before they can attempt the material of the course. In all three course outlines, there were recommendations for the pre-requisite courses students were expected to have successfully passed before enrolling in the introductory course. This is well supported by studies on how prepared students are for introductory and advanced circuits courses. For example, the work of [54] highlights that students are often exposed to circuit concepts in physics and other science courses. However, for prior knowledge to be beneficial to learning new and related material, this prior knowledge has to be both accurate and adequate [55]. The observations and interviews with all the professors further supported this point, as the professors all discussed that there seemed to be an apparent mismatch between what the students should know based on their prior classes and how they interacted with the material in the course. It was discussed in the interviews and repeatedly stated in the observations that there was a certain level of mathematical skill required to be successful in understanding the material related to circuits. However, all three professors discussed that a primary reason for the difficulty encountered by students on the course was their lack of sufficient prior mathematical knowledge.

The various categories of students based on their abilities were discussed as being influential on the types of examples they would present in class or the kind of questions they would ask, even as far as how they designed quiz and exam items. Deep learning, as defined by the literature and in this class, is marked by the ability to transfer knowledge from one context to another [56–58]. In fact, knowledge transfer is a critical and important aspect of learning in any context. The work of [59] highlights that educational learning environments should be designed with knowledge transfer as an intended outcome. In this particular study, knowledge transfer is particularly crucial, since this course is foundational to the rest of the electrical engineering degree program. Two of the professors discussed that it was not a case of the students lacking the capability to understand the concepts covered in class, it was more a case of them not being able to take that knowledge and apply it to problems of a different nature. The importance of knowledge transfer was also explicit in two of the three course documents and was reinforced in the classes. Time was spent at the beginning of a class in two sections to emphasize the concepts that were important for exams or achieving good grades on upcoming tests.

### 4.4.2. Characteristics of Content

In addition to the perceived characteristics of the students, the nature of the content was found to be a determining factor in how the course concepts were taught to the students.

The field of electrical engineering is inherently abstract and highly technical. In all three units, the abstract nature of the concepts was discussed and presented as an area that tends to be problematic. This was evident in the interviews, as professors would individually speak to the fact that the concepts are abstract and hence reinforce the need for strong mathematical skills. As noted in the Background section, advanced mathematical thinking is a requisite skill for circuit analysis. This is motivated by the fact that circuit analysis involves intense mathematical manipulation [60]. In unit one and two, however, this heavy emphasis on mathematical representation was more explicit. For example, when a student would ask a clarifying question, or in situations where professors would pose a question to the students, the explanation or justification of the given response would be prefaced by the importance of having certain mathematical skills, followed by the derivation of equations and other mathematical notation. In unit three, however, student questions would be answered using multiple representations, such as a graphic illustration or a qualitative discussion, in which the professor would make connections between various sections of the circuit or the problem being solved. Multiple representation has been recommended as an important feature in learning complex and abstract concepts. According to [61], the use of multiple representations when learning scientifically complex concepts can support the construction of a deeper understanding of the content.

Through all the sources of the data collected, it was observed in unit three that, while mathematical knowledge and skills were important for understanding the content of the class, they were not the only means by which the course content could be conveyed. In all three units, the fact that the content covered in this course was important for future courses was stressed repeatedly. All three professors discussed in the interviews and throughout the class observations that, while the concepts seemed disjointed now, they would become applicable as the students moved on to more advanced classes. It was also discussed in the interviews that the content of this course was meant to give students a broad overview of all the possible areas of study. Hence, the way the course is taught is more applicable to electrical engineering majors, since their choice is to continue a career in this discipline.

### 4.4.3. Structured Problem-Solving Process

The course objectives, as presented in the course catalog and outline, are to: "1. analyze linear resistive circuits, 2. analyze 1st order linear circuits with source and/or passive elements and 3. analyze 2nd order linear circuits with sources and/or passive elements" (course outline p. 4). In order to help the students to achieve these outcomes, the professors discussed the use of class examples aimed at helping the students realize the importance of "applying a problem-solving approach much like they would follow a recipe in a cooking class" (professor of unit one), "being able to apply what tools they have on their belts" (professor of unit two), or "problem-solving skill development" (professor of unit three). This indicates that there is an emphasis on students being able to master the process of solving circuit problems that could be of a clinical or real-life nature.

In addition to being able to apply this problem-solving process, the importance of repetitive practice and developing a deep understanding of the methods of solution were skills the students were encouraged to acquire. Researchers posit that deliberate practice not only enhances students' problem-solving accuracy and speed, but also leads to increased student motivation. In fact, one key benefit of repeated deliberate practice is that the more a student engages with the activity, their performance increases, which, in turn, motivates them to learn more [37,62]. All three professors discussed that, in their own learning of these concepts, they developed an attitude of repetitive practice, which they found to be helpful for learning the difficult material. Providing opportunities for repetitive practice and making students aware of the understanding of its importance were most explicit in the course documents; in all three course outlines, the importance of completing homework as a method of repetitive practice was expressed. For example: unit one and two had the following statements "The homework is a very important part of the course. You may read your lecture notes and the text and think that you understand the material. However,

when you attempt to work the homework problems, you will frequently find that you did not…you are strongly advised to solve independently as much of the homework as you possibly can. This will serve you well come exam time…to avoid having to memorize formulas, we will provide you some formulas for your use during the exam". In unit three, the wording was different, but the concept remained the same: "work the homework: The problem-solving techniques taught in this class are as important as the theoretical material. Memorizing formulas and understanding the concepts will not be enough to pass the class. You will only learn how to solve the problems if you work the homework".

The importance of this practiced problem solving was reinforced during the teaching of the content in the classes. In unit three, the professor would draw boxes or use asterisks to emphasize equations or solution processes that were important for the students to learn. Additionally, it was observed that, at the beginning of the classes, the professors would emphasize which concepts were the most important to become familiar with, as well as a measure of how these concepts related to upcoming tests, exams, or future concepts to be taught or explored by the students.

### 4.4.4. Student Engagement

The issue of engaging students was discussed by all the professors as being hard to achieve, especially with non-electrical engineering majors, for two main reasons. The first reason was the inability of students to apply the material of this course to their other coursework. The professors explained that, while mechanical engineering students (a reference made by all three professors) have some kind of tangible entity with which to associate what they learn in mechanical introductory courses, when they come to this circuit course, they tend to sometimes feel inept, because there is nothing to relate the concepts to. The second reason relates to the fact that the course is compulsory; hence, students take the course out of obligation, and less due to being motivated to do so. This, they discussed, works to the detriment of their efforts to engage students. The professor of unit three discussed extra-credit activities he added to the course as a means of getting students involved in the class. However, none of the students expressed an interest. In addition, considering the fact that this study was conducted in the off-semester for EE majors, it could be possible that the lack of engagement observed was a result of the other engineering majors that were also enrolled in the course at the time.

The use of analogical comparisons to real-world concepts such as talking on cell phones, playing sports or music, and following a cooking recipe were examples used in the units to make the concepts relatable. The use of analogies and comparative language in learning is a common practice, especially when teaching circuits. Several researchers have recommended the use of analogies to make the unknown known. However, these researchers also cautioned against the overuse of analogies, as they can lead to misunderstandings [63,64]. All the professors discussed the importance of using analogies, but also the limitations associated with their use. Analogies were therefore described as appropriate methods for first introducing concepts, but over time, the professors hoped that students understood the concepts on their own, thus rendering the analogies no longer useful. The fact that there is a point at which the analogy will break down or become incapable of being exactly aligned with the concept being taught was discussed as a limitation and reason for opting not to use analogies, as can be seen from the interview quotes presented earlier.

The use of instructional technology created an instructor-focused atmosphere that was most prominent in units one and two. In these two classes, the professors lectured from pre-made slides and would use a digital tablet to solve problems or expand on equation derivations. This was observed as a limitation in how the professors could interact with the class, as they were basically tethered to the lectern in the room. In addition, students were given handouts with slides for the class at the beginning of the session. This significantly reduced their participation in the class. The student engagement determined by the observation protocol reflected this fact. In unit three, however, the professor wrote on the chalkboard, except when having conversations with the students. It was observed

that the students in unit three were more attentive to the professor, asked more questions, and responded to his questions. This could be attributed to the fact that they were having to pay attention to the lecture and make their own notes. The professor discussed this as a strategy he enforced in his class to not only engage the students to pay attention, but to encourage them to think about the concepts being discussed during class, and even after class when he would release his own lecture notes.

4.4.5. Impact of Time Constraint

The emerging themes informing this pattern were primarily found across in the data collected from the professor interviews. The professors discussed, when asked about the challenges faced when teaching the concepts of this course, restrictions associated with time as one of the main challenges. Decisions such as how much content should be covered in one session, the depth of explanations, discussions on alternative problem-solving strategies, exploring more advanced content, reinforcing what was covered in class, and "striking a balance between concept and worked problems" were some of the most commonly discussed issues. The influence of time constraints was an important finding, in that even though the professors wished to spend more time reinforcing concepts or using more real-life applications, the design of the course did not afford them this opportunity. The nature of the course and by extension the lecture classes are constructed to expose students to a wide variety of topics. Consequently, there is not a lot of time spent in the course on going deeper into concepts or spending time ensuring students are exposed to multiple problem-solving strategies. The rigidity of the engineering curriculum is a common area of concern for researchers, in that there is often so much content to cover that there is very little room for exploration [7,65].

The professors of unit one and three also talked about how they went about working on problems in class which were very different from those problems the students encountered in the exams. This is because, within the 50 min session, it is expected that the relevant concepts for that session will be covered, and some example problems will be worked out. In addition, one of the professors discussed that, even though it was obvious to him that the students could benefit from having additional discussions and explanations about the concept, "it would not be fair to spend more time of these concepts as there is a risk of the class falling behind based on the collective class schedules". A coping mechanism discussed by two of the professors involved suggesting to interested students other projects they could attempt on their own to obtain a deeper experience with the concepts. However, the professors acknowledged that this approach only worked for the higher-performing students.

The findings indicate that there are various intervening factors that helped to determine how this introductory course was designed and the concepts were taught to the students. Through the multiple sources collected from each unit, it was found that the main emphasis of this course was providing students with the necessary information to develop a structured problem-solving process that could be applied to different circuit configurations. It was also found that repeated practice was strongly encouraged as a means of ensuring the students not only understood the content of the class, but most importantly for achieving a good grade on exams. A third major finding was the heavy reliance on mathematical knowledge and skills that was deemed necessary if the students were going to be successful on the course. All three professors discussed that the students seemingly lacked this prior knowledge, or their understanding of these very core concepts was insufficient. This perceived inadequate prior knowledge is problematic when professors attempt to use mathematical representation to convey information about the abstract concepts covered in the class.

## 5. Implications and Limitations of the Study

The implication of this study is centered on the design of learning environments and the decisions made about how to teach and assess students. The issue of time constraints

and having to keep to a rigid schedule was an important finding in this study. This indicated the need for an investigation into the number of topics being covered on the course and how the relationships between these concepts can be leveraged. This would ensure that students are exposed to all the necessary information, while still being able to acquire the relevant skills necessary to move forward in their course of study. This work also has implications for how concepts are taught. A balanced approach that utilizes multiple representations of concepts [61] can have a significant impact on students' overall learning.

This study also highlighted the need for connections between the theoretical mathematical principles taught in a calculus course to the application of these principles as needed in an engineering course [41]. The findings of this study indicated that, in some cases students were enrolled in the calculus course at the same time as this circuits' analysis course. This is potentially problematic, as the students were perceived as lacking the requisite mathematical knowledge, which, in turn, hindered their ability to grasp the circuit analysis concepts needed to be successful in the course [55].

This work, like all research projects, has its own set of limitations. Firstly, only three sections of the same course at one university were studied. Secondly, only two topics of the entire course were observed, though the professor interviews attempted to capture their experiences over the duration of the course. Thirdly, all the data collected were from the perspective of the instructor. Future studies should consider including the students' perspective of how they learn these concepts. This could be conducted through focus groups or one-on-one student interviews. Finally, this study did not include data from textbooks, though they play a major role in most courses and related assessments. The decision not to include textbooks was based on the focus of the study on the course design decisions made by the instructors; hence, only materials created and used by the instructors were examined.

## 6. Conclusions

The goal of this study was to describe how introductory circuit concepts are taught to engineering students enrolled in a compulsory sophomore-level course. To explore this phenomenon, a descriptive case study with embedded units was conducted. Significant insights were gained from this study in terms of how professors used their content knowledge, experiences with teaching, and their own experiences of learning these concepts to not only design, but structure their courses. Studies of how students learn abstract and complex concepts all indicate that their prior knowledge, repeated practice, and use of analogical reasoning are important factors that should be taken into consideration. This work holds several implications for educators who are keen on designing and teaching courses that are foundational in nature. With a holistic analysis of the course documents, in-class observations, and semi-structured interviews, themes that pertain to the nature of the content, students' characteristics in terms of their background knowledge and mathematical understanding, and the decisions made about how to present content, this work provides insights into the significant role each aspect plays in students learning. These initial understandings could be further developed and explored to highlight the critical roles the overall design of a course and the delivery of its content can play in introductory courses.

**Funding:** This research received no external funding.

**Institutional Review Board Statement:** The study was conducted in accordance with the Declaration of Helsinki, and approved by the Institutional Review Board (or Ethics Committee) of Virginia Tech (protocol code 17-1106 and 29 November 2017).

**Informed Consent Statement:** Informed consent was obtained from all subjects involved in the study.

**Data Availability Statement:** The data used and analyzed in this study are available from the corresponding author upon reasonable request.

**Conflicts of Interest:** The author declares no conflict of interest.

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
