# Peer review of "Teaching Complex Introductory Concepts in a Sophomore Circuits Course: A Descriptive Case Study"

_education, doi:10.3390/educsci13101022_

Round 1

Reviewer 1 Report

The authors present a good contextualization for the study, review relevant literature, and provide a comprehensive study dealing with various issues. The research questions are clearly stated

 However, in the background, they focus strongly on previous knowledge and do not address other issues relevant to analyzing data (more critically). Examples of such issues are problem-solving (and how it opposes problem-based learning) which would help to understand teachers’ practices emphasis repetition; the difference between physics and mathematics which would enable authors to go further on data discussion regarding the professors’ strong emphasis on the mathematical part of concepts. The concept of assessment, together with the match between school-based assessment, external (national) assessment, and course-learning outcomes should deserve some attention as well. These issues would enable a deeper discussion on the teacher’s practices and (mis)worries with regard to preparing students for the external evaluation.

There some lack is of information on data collection instruments namely with regard to their origin, structure, as well as to their development, and validation processes.

The findings section is too long and has scarce empirical support. Probably, the authors would do better to select only a part of the issues to analyze in this manuscript so that they could provide empirical data (transcripts) to support the interpretations they put forward. A consequence of this would be that the title and the research questions would need to be adjusted.

Information on ethical issues /ethical approval of the project should be provided.

Relevant literature should be used explicitly for data discussion or in the conclusion section to place the findings within the state of the art.

The implications of the results are scarcely discussed. Authors should develop the discussion on this issue mentioning aspects in need of improvement and giving insights on how they may be improved.

Finally, the limitations of the study should be discussed for the implications they may have for the reliability of the findings.

Author Response

The authors present a good contextualization for the study, review relevant literature, and provide a comprehensive study dealing with various issues. The research questions are clearly stated

  • Thank you for your kind comments. I appreciate the time taken to perform this review and the useful suggestions provided below.

 However, in the background, they focus strongly on previous knowledge and do not address other issues relevant to analyzing data (more critically). Examples of such issues are problem-solving (and how it opposes problem-based learning) which would help to understand teachers’ practices emphasis repetition; the difference between physics and mathematics which would enable authors to go further on data discussion regarding the professors’ strong emphasis on the mathematical part of concepts. The concept of assessment, together with the match between school-based assessment, external (national) assessment, and course-learning outcomes should deserve some attention as well. These issues would enable a deeper discussion on the teacher’s practices and (mis)worries with regard to preparing students for the external evaluation.

  • Thank you for this suggestion. The background section was edited to add new literature that aligns with the issues highlighted by this reviewer.

There some lack is of information on data collection instruments namely with regard to their origin, structure, as well as to their development, and validation processes.

  • Thank you for pointing this out. I have made some additions to the description of the observations and interviews to explain how the protocols were developed. I also included the IRB number which will be uncovered after the review process.

The findings section is too long and has scarce empirical support. Probably, the authors would do better to select only a part of the issues to analyze in this manuscript so that they could provide empirical data (transcripts) to support the interpretations they put forward. A consequence of this would be that the title and the research questions would need to be adjusted.

  • The findings section was shortened to highlight more concise information that directly relates to the research questions.

Information on ethical issues /ethical approval of the project should be provided.

  • The IRB number will be included in the finalized version of the manuscript.

Relevant literature should be used explicitly for data discussion or in the conclusion section to place the findings within the state of the art.

  • The background section was revised to add more research that was also aligned with the discussion section of the manuscript.

The implications of the results are scarcely discussed. Authors should develop the discussion on this issue mentioning aspects in need of improvement and giving insights on how they may be improved.

  • An implications section was added to the manuscript  

Finally, the limitations of the study should be discussed for the implications they may have for the reliability of the findings.

  • A limitations section was added to the manuscript.

Reviewer 2 Report

Interesting paper - excellent use of literature

Is the term 'complex' needed?

Abstract can be strengthened by including important observations offered in Conclusion section.

Surprised that textbooks or other collateral materials were not part of the study.

Interesting observation on distinction between EE and ME majors - why is the preprofessional identities so strong already at sophomore level? Does requiring lab for EE majors only signal other majors that less engagement is expected - reinforcing premature attitudes among majors?

Check referencing - e.g., p554 at line 216, Fig. 5.3 vs (correct( Fig. 3 at line 358.

Author Response

Interesting paper - excellent use of literature

  • Thank you for your kind comments.

Is the term 'complex' needed?

  • The use of the term complex was clarified and the title of the manuscript was revised.

Abstract can be strengthened by including important observations offered in Conclusion section.

  • Thank you for this suggestion. This edit was made to the abstract.

Surprised that textbooks or other collateral materials were not part of the study.

  • Thank you for this query. Since the study was seeking to uncover the instructional strategies used by the instructor and how their belief of how their students learn factor into how they design the courses the decision was made to only explore the course materials directly developed by the instructors. Though I do recognize that these instructors also use the textbooks for their notes and class homework. I have made a note of the lack of inclusion of textbook materials in the limitation section.

Interesting observation on distinction between EE and ME majors - why is the preprofessional identities so strong already at sophomore level? Does requiring lab for EE majors only signal other majors that less engagement is expected - reinforcing premature attitudes among majors?

  • Great question. While the course is compulsory for all majors, it only serves as a pre-requisite for other courses taken by the EE students. For the other majors, it is more of a checking the box kind of situation. This might explain their lack of engagement to some degree.

Check referencing - e.g., p554 at line 216, Fig. 5.3 vs (correct( Fig. 3 at line 358.

  • Thank you for this correction. The manuscript was also thoroughly proof read to check for any other errors such as these.

Round 2

Reviewer 1 Report

The authors have responded properly to almost all the comments and the manuscript has improved accordingly. The weakness of the lack of explicit use of literature improved a bit. Still, it could have improved more (no reference is mentioned in the implications and limitations as well in the conclusions section).

Even though I am not a native speaker of English, the English language needs light editing because there are several basic grammatical mistakes that should be corrected before publication.

Author Response

Thank you for the time taken to review this manuscript again. Your comments and recommendations are appreciated. The following changes were made to address the thoughtful comments received. 

  1. References were added to support the implications of the study. However, there were no references added to the limitations and conclusions portions of the manuscript in keeping with the practice of engineering education and broadly social science research not to add references to the limitations or conclusion sections.
  2. A thorough proof reading of the manuscript was conducted to revise and edit grammatical and other subject verb disagreements where they occurred.